# Mitochondrial Dysfunction in Podocytes Caused by CRIF1 Deficiency Leads to Progressive Albuminuria and Glomerular Sclerosis in Mice

**DOI:** 10.3390/ijms22094827

**Published:** 2021-05-02

**Authors:** Ki Ryang Na, Jin Young Jeong, Jin Ah Shin, Yoon-Kyung Chang, Kwang-Sun Suh, Kang Wook Lee, Dae Eun Choi

**Affiliations:** 1Department of Nephrology, Chungnam National University School of Medicine, Daejeon 35015, Korea; drngr@cnu.ac.kr (K.R.N.); spwlsdud@naver.com (J.Y.J.); wlsdkahh@gmail.com (J.A.S.); 2Department of Medical Science, Chungnam National University School of Medicine, Daejeon 35015, Korea; 3Department of Nephrology, Daejeon St. Mary’s Hospital, Catholic University of Korea, Daejeon 34572, Korea; racer@catholic.ac.kr; 4Department of Pathology, Chungnam National University School of Medicine, Daejeon 35015, Korea; kssuh@cnu.ac.kr

**Keywords:** CRIF1, mitochondrial oxidative phosphorylation, podocyte, albuminuria, glomerular sclerosis

## Abstract

Recent studies have implicated mitochondrial disruption in podocyte dysfunction, which is a characteristic feature of primary and diabetic glomerular diseases. However, the mechanisms by which primary mitochondrial dysfunction in podocytes affects glomerular renal diseases are currently unknown. To investigate the role of mitochondrial oxidative phosphorylation (OxPhos) in podocyte dysfunction, glomerular function was examined in mice carrying a loss of function mutation of the gene encoding CR6-interacting factor-1 (CRIF1), which is essential for intramitochondrial production and the subsequent insertion of OxPhos polypeptides into the inner mitochondrial membrane. Homozygotic deficiency of CRIF1 in podocytes resulted in profound and progressive albuminuria from 3 weeks of age; the CRIF1-deficient mice also developed glomerular and tubulointerstitial lesions by 10 weeks of age. Furthermore, marked glomerular sclerosis and interstitial fibrosis were observed in homozygous CRIF1-deficient mice at 20 weeks of age. In cultured mouse podocytes, loss of CRIF1 resulted in OxPhos dysfunction and marked loss or abnormal aggregation of F-actin. These findings indicate that the OxPhos status determines the integrity of podocytes and their ability to maintain a tight barrier and control albuminuria. Analyses of the glomerular function of the podocyte-specific primary OxPhos dysfunction model mice demonstrate a link between podocyte mitochondrial dysfunction, progressive glomerular sclerosis, and tubulointerstitial diseases.

## 1. Introduction

Mitochondria are essential intracellular organelles that have a major role in energy production via ATP synthesis [1,2]. Patients with mitochondrial dysfunction caused by inherited mutations of mitochondrial DNA (mtDNA) display renal manifestations such as proteinuria and glomerular sclerosis [3,4,5]. Podocytes are specialized epithelial cells that maintain the critical glomerular filtration barrier [6,7] and their dysfunction is frequently associated with glomerular proteinuria, which leads to end-stage renal disease [8]. Progressive renal diseases, including diabetic kidney disease and glomerulonephritis, are associated with both mitochondrial dysfunction and podocyte injury [9,10]. Although patients with mtDNA mutations develop renal diseases with characteristic glomerular sclerosis [11], it is unclear how mitochondrial dysfunction in the kidney leads to glomerular diseases. Although podocytes contain a relatively small number of mitochondria, the specialized barrier function of these cells, which is maintained by their actin cytoskeleton, may be influenced by the oxidative phosphorylation (OxPhos) capacity of the mitochondria [6,12]. However, the functional relationship between mitochondria and the actin cytoskeleton in podocytes is not fully understood. 

CRIF1 plays an essential role in mitochondrial synthesis and membrane integration of OxPhos polypeptides [13]. CRIF1 interacts with proteins surrounding the polypeptide exit tunnel of the large subunit of mitochondrial ribosomes and mediates the integration of nascent OxPhos polypeptides into the inner mitochondrial membrane with the assistance of the specific chaperone Tid1. Deletion of the *Crif1* gene results in a marked failure of OxPhos both in vitro and in vivo, and tissue-specific CRIF1-deficient mice undergo distinct organ degeneration [13,14,15]. To investigate the renal manifestation of mitochondrial dysfunction, we developed mice with podocyte-specific mitochondrial OxPhos dysfunction by deleting *Crif1* using conditional gene targeting.

## 2. Results

### 2.1. CRIF1^pdKO^ Mice Showed Massive Albuminuria and Progressive Renal Failure

CRIF1*^flox/flox^* mice were crossed with podocin-*Cre* mice, in which *Cre*-recombinase expression is directed to podocytes starting from the capillary loop stage during glomerular development [16]. Quantitative real-time PCR analyses of genomic DNA prepared from tail tissue were performed to verify homozygous and heterozygous deletions of the *Crif1* gene (Figure 1a). The renal functions of homozygous CRIF1-deficient CRIF1*^flox/flox^* with podocin-*Cre* (CRIF1^pdKO^) mice were compared with those of their gender-matched CRIF1*^flox/flox^* without podocin-*Cre* (CRIF1^ctrl^) littermates. The female CRIF1^pdKO^ mice developed normally until 20 weeks of age, after which their body weights were lower than those of the CRIF1^ctrl^ mice. By contrast, the body weights of the male mice were unaffected by deletion of *Crif1* (Figure 1b). All CRIF1^ctrl^ mice survived up to 80 weeks, whereas the survival rate of the CRIF1^pdKO^ mice dropped after 25 weeks of age and all CRIF1^pdKO^ mice died within approximately 38 weeks (Figure 1c). Consistent with the increased mortality, the blood urea nitrogen and serum creatinine levels were significantly higher in the CRIF1^pdKO^ mice than the CRIF1^ctrl^ mice after 10 weeks of age, suggesting a progressive deterioration of glomerular function following knockout of *Crif1* (Figure 1d,e). A progressive increase in the level of albuminuria also occurred in the CRIF1^pdKO^ mice from 3 weeks of age (Figure 1f). Taken together, these results indicate that loss of CRIF1 in podocytes results in albuminuria and renal dysfunction.

### 2.2. CRIF1^pdKO^ Mice Showed Progressive Glomerular Sclerosis and Tubular Fibrosis

Renal cortex sections from CRIF1^pdKO^ and CRIF1^ctrl^ mice were also stained with hematoxylin and eosin or Masson’s Trichrome stain. No differences between the two groups were observed at 3 or 5 weeks of age (Figure 2). At 10 weeks of age, unlike the CRIF1^ctrl^ mice, the CRIF1^pdKO^ mice developed focal segmental glomerular sclerosis (FSGS) (Figure 2; yellow arrows), tubules filled with proteinaceous casts (Figure 2; black arrows) and mild-to-moderate interstitial inflammation and fibrosis (Figure 2). At 20 weeks of age, the CRIF1^pdKO^ mice showed prominent global glomerular sclerosis (Figure 2; yellow arrows), proteinaceous casts (Figure 2; black arrows), tubular detachments, tubular epithelial thinning, and severe interstitial inflammation associated with fibrosis (Figure 2 and Appendix A). At 10 and 20 weeks of age, the levels of tubulointerstitial fibrosis and glomerulosclerosis were significantly higher in the CRIF1^pdKO^ mice than the CRIF1^ctrl^ mice (Figure 2).

### 2.3. CRIF1^pdKO^ Mice Showed Progressive Podocyte Injury

Transmission electron microscopy analyses showed that, although 3-week-old CRIF1^pdKO^ mice had normal podocytes and foot processes, they displayed mitochondrial structural abnormalities, including swelling, abnormal arrangement, and loss of cristae (Figure 3). At 5 weeks of age, mild foot process effacement occurred in CRIF1^pdKO^ mice (Figure 3; yellow arrows) and the foot process width was significantly higher in CRIF1^pdKO^ mice than CRIF1^ctrl^ mice (Figure 3). Foot process fusion and effacement were also observed in the 10-week-old CRIF1^pdKO^ mice (Figure 3; yellow arrows). At 20 weeks of age, the foot processes in CRIF1^pdKO^ mice showed extensive effacement and microvillous transformation (Figure 3; red arrows), and podocyte degeneration and vacuolization also occurred (Figure 3; black arrows). Overall, podocyte-specific, CRIF1-deficient mice showed mitochondrial abnormalities and foot process defects and went on to develop progressive global glomerular sclerosis associated with tubulointerstitial fibrosis, which leads to renal failure.

### 2.4. Podocyt-Specific CRIF1 Deletion Causes Structural Abnormality of Mitochondria

Next, transmission electron microscopy was used to examine ultrastructural changes in the mitochondria and foot processes of podocytes from 10-week-old CRIF1^pdKO^ and CRIF1^ctrl^ mice. Compared with CRIF1^ctrl^ mice, CRIF1^pdKO^ mice displayed a marked loss of foot processes and an accumulation of abnormal mitochondria (Figure 4A). The mitochondrial cristae in podocytes from CRIF1^ctrl^ mice were densely packed, whilst those in podocytes from CRIF1^pdKO^ mice were swollen and less dense (Figure 4B). These findings suggest that CRIF1 is required to maintain normal mitochondrial structure in podocytes.

### 2.5. CRIF1 Deletion of Podocyte Leads to Mitochondrial Dysfunction

Immunofluorescent staining of endogenous CRIF1 in immortalized mouse podocytes overlapped mitochondrial staining, suggesting that CRIF1 is localized exclusively to the mitochondria of podocytes (Figure 5A). Compared with podocytes transfected with a control siRNA, those transfected with a CRIF1-specific siRNA had significantly lower levels of CRIF1 protein and *Crif1* mRNA (Figure 5B), significantly lower levels of OxPhos subunits in the mitochondrial protein complex IV (Figure 5C), and lower oxygen consumption rates (OCRs) (Figure 5D). Basal oxygen consumption in CRIF1-specific siRNA-transfected cells was 68% of that in control siRNA-transfected podocytes (*p* < 0.05 by Student’s *t*-test, Figure 5d). The maximal mitochondrial respiration capacity, evaluated by treatment of the podocytes with the electron transport chain decoupler carbonyl cyanide m-chlorophenylhydrazone (CCCP), was lower in CRIF1-specific, siRNA-transfected podocytes than control podocytes (Figure 5d). These findings indicate that CRIF1 is required for mitochondrial OxPhos in cultured podocytes.

### 2.6. CRIF1 Deletion of Podocyte Leads to Loss and Aggregation of F-Actin

To determine whether CRIF1-deletion-induced mitochondrial dysfunction affects the cytoskeleton of the podocyte slit diaphragm, the expression levels of the cytoskeletal proteins F-actin, α-actinin-4 and synaptopodin, and the slit diaphragm proteins nephrin, cofilin and ZO-1, were examined in siRNA-transfected podocytes by immunoblotting. F-actin expression was significantly lower in CRIF1-specific siRNA-transfected podocytes than control siRNA-transfected podocytes; however, the expression levels of the other proteins examined were not affected by knockdown of CRIF1 (Figure 6A). Immunofluorescent staining of phalloidin showed that control podocytes expressed F-actin in longitudinal stress fibers, whilst CRIF1-specific siRNA-transfected podocytes showed a marked loss of expression and dot-like aggregation of F-actin (Figure 6B). The size of podocytes was similar between scRNA-transfected podocytes and CRIF1-specific siRNA-transfected podocytes (Appendix A). At 10 weeks of age, glomerular nephrin distribution was comparable in CRIF1^pdKO^ and CRIF1^ctrl^ mice; however, the merged F-actin and nephrin-stained area was significantly smaller in CRIF1^pdKO^ mice than CRIF1^ctrl^ mice (Figure 6C).

## 3. Discussion

In this study, we first demonstrated the role of mitochondria in the maintenance of podocyte structure and function using a podocyte-specific mitochondrial dysfunction model. Podocyte-specific CRIF1 deletion leads to mitochondrial dysfunction and structural abnormalities, causing massive albuminuria and progressive glomerulosclerosis in mice.

Podocytes contain a high number of mitochondria, which provides a high energy supply to maintain various cellular functions, including the organization of cytoskeletal and extracellular matrix proteins [3]. Drugs such as puromycin and aristolochic acid cause mitochondria and podocyte dysfunction, leading to profuse proteinuria [17,18,19]. In animal models of renal disease, including streptozotocin-induced diabetic rats, renal hypertension is frequently associated with mitochondrial defects in podocytes [9,20]. Although these observations suggest that mitochondrial OxPhos function is closely related to podocyte dysfunction and albuminuria, the mechanisms by which primary mitochondrial dysfunction in podocytes causes defective barrier function are currently unknown. The mouse model of podocyte-specific CRIF1 deficiency described here will be useful to identify the role of primary OxPhos dysfunction in podocytes during albuminuria and renal disease progression. Notably, the podocyte-specific, CRIF1-deficient mice showed high levels of albuminuria and progressive segmental glomerulosclerosis. Mitochondrial cytopathy resulting from mtDNA mutations or deletions may cause glomerular disease characterized by sclerotic changes similar to FSGS [11]. Mitochondrial cytopathy caused by mtDNA mutation involves all the cell components forming glomeruli [21]. Although podocytes contain a relatively low density of mitochondria compared with tubular cells, they are susceptible to mitochondrial cytopathy [4,22,23]. The results presented here suggest that podocyte-specific mitochondrial OxPhos dysfunction is a plausible underlying cause of progressive FSGS-like syndrome. The CRIF1-deficient mice also underwent tubulointerstitial changes characterized by multiple proteinaceous casts, interstitial inflammation and fibrosis, which are secondary outcomes of prolonged and marked albuminuria. 

Derangement of the cytoskeletal structure is the common pathway of loss of the specialized barrier function of podocytes. For example, mitochondrial injury induced by high glucose or puromycin treatment rearranges F-actin alignment and changes podocyte shape, which may cause foot process abnormalities and proteinuria [24,25]. The finding that defects in the regulation of mitochondrial dynamics cause effacement of foot processes suggests that normal maintenance of the cytoskeletal integrity requires intact mitochondrial function [26]. Here, podocyte-specific mitochondrial dysfunction caused by knockout of CRIF1 reduced F-actin expression and disorganized its arrangement in cultured podocytes. These findings support the concept that mitochondrial OxPhos dysfunction affects the actin cytoskeleton, leading to the disruption of podocyte architecture and the tight glomerular barrier. Mitochondrial dysfunction induced by CRIF1 depletion may preferentially affect F-actin arrangement.

## 4. Materials and Methods

### 4.1. Mice

Floxed CRIF1 (CRIF1*^flox/flox^*) mice were generated as described previously [15]. The podocin-*Cre* transgenic mice were kindly provided by Dr. Susan Quaggin (Mount Sinai Hospital, Toronto, Canada). The animals were maintained in the animal colony at Chungnam National University School of Medicine (Daejeon, Korea) under the institutional guidelines of the Korean Research Institute of Biotechnology and Bioscience and were cared for according to a protocol approved by the Chungnam National University Institutional Animal Care and Use Committee (CNU-00456). Podocin-*Cre*:CRIF1*^flox/flox^* mice (CRIF1^pdKO^) were generated by mating CRIF1*^flox/flox^* mice with podocin-*Cre*:CRIF1*^flox/+^* mice. Sex-matched littermates with a genotype of CRIF1*^flox/flox^* without podocin-*Cre* were used as controls (CRIF1^ctrl^).

### 4.2. Blood and Urinary Measurements

The body weights of the mice were measured once every two weeks. Urinary albumin excretion was measured using an Albumin (Mouse) Elisa Kit (ALPCO^®^, Salem, NH, USA), according to the manufacturer’s instructions. Blood samples obtained at the time of death were used to determine serum creatinine and blood urea nitrogen levels using chemistry autoanalyzer, Toshiba 200FR (Toshiba Medical Systems Co., Tokyo, Japan).

### 4.3. Histological Evaluation

Paraffin-embedded kidneys were cut into 4 μm sections and mounted onto glass slides. The sections were deparaffinized with xylene, stained with hematoxylin and eosin or Masson’s Trichrome stain, and then examined under a BX51 microscope (Olympus, Tokyo, Japan). The severity of glomerular injury (glomerulosclerosis score) was determined as follows: 0, no sclerotic changes in the glomerulus; 1, <25% sclerosis; 2, 25–50% sclerosis; 3, 50–75% sclerosis; 4, >75% sclerosis. Thirty glomeruli were randomly selected in each tissue section and the average score was calculated. The severity of tubulointerstitial injury was evaluated by examining ten fields in randomly selected tissue samples stained with Masson’s Trichrome stain, and the results were expressed as a percentage of the relative area of the blue stained- and interstitial areas. All images were quantified using Image-Pro^®®^ Plus version 6.0 software (Media Cybernetics, Silver Spring, MD, USA).

### 4.4. Transmission Electron Microscopy

Transmission electron microscopy was performed as described previously [15]. Briefly, tissues were fixed with 1% glutaraldehyde in phosphate-buffered saline (PBS) for 2 h at 4 °C, and then embedded in EMbed-812 resin (Electron Microscopy Sciences, Hatfield, PA, USA). The samples were observed using a transmission electron microscope (H-7650; Hitachi, Japan). The foot process width was assessed by measuring the number of filtration slit membranes per 100 mm of glomerular basement membrane.

### 4.5. Immunostaining and Confocal Microscopy

Immunostaining and confocal microscopy were performed as described previously [15]. Briefly, goat anti-human CRIF1 antibodies (sc-103443 and sc-103445; Santa Cruz Biotechnology, Dallas, TX, USA) were used to detect mouse CRIF1 and an anti-Tom20 antibody (#612278; BD Biosciences, San Jose, CA, USA) was used to identify the mitochondria. For immunostaining, podocytes were fixed with 4% paraformaldehyde for 15 min, permeabilized in PBS containing 0.2% Triton X-100 (Sigma-Aldrich, St. Louis, MO, USA), and then blocked with blocking buffer (10% horse serum, 1% BSA, 0.002% NaN_3_, and 1 × PBS) for 45 min at room temperature. Podocytes and 5 μm kidney cryostat sections were incubated with primary and secondary antibodies diluted in PBS containing 1% BSA at 4 °C overnight and room temperature for 45 min, respectively. The specimens were visualized using an LSM 510 META confocal microscope (Carl Zeiss AG, Oberkochen, Germany). All images were quantified using Image-Pro^®®^ Plus version 6.0 software (Media Cybernetics, Silver Spring, MD, USA).

### 4.6. Podocyte Culture and siRNA Transfection

Conditionally immortalized mouse podocytes were cultured as described previously [27,28,29]. The podocytes were maintained in RPMI 1640 medium (Gibco-BRL, Eggenstein, Germany) supplemented with 10% fetal bovine serum and antibiotics. For propagation, the podocytes were cultured on type I collagen (Sigma-Aldrich) at 33 °C and the culture medium was supplemented with 10 U/mL recombinant interferon-γ to enhance T-antigen expression. To induce differentiation, the podocytes were maintained in DMEM (Gibco-BRL, Carlsbad, CA, USA) supplemented with 5% fetal bovine serum on type I collagen at 37 °C without interferon-γ. Differentiation of podocytes grown for 14 days at 37 °C was confirmed by immunocytochemical staining for synaptopodin, a podocyte differentiation marker. The podocytes were grown to near confluency and were serum-deprived for 24 h prior to use. Commercially available CRIF1-specific siRNAs (Invitrogen, Groningen, The Netherlands) were transfected into podocytes using Lipofectamine 2000 Reagent (Invitrogen). Briefly, each siRNA (80 μM) was mixed with Lipofectamine in Opti-MEM I medium (Gibco-BRL, Carlsbad, CA, USA). Each siRNA-Lipofectamine mixture was then added to the cells and incubated overnight at 37 °C and 5% CO_2_. After washing, the podocytes were incubated with normal culture medium. The cells were harvested for protein extraction 72 h after transfection.

### 4.7. OCR

Measurement of the mitochondrial OCR was performed as described previously [27], using a Seahorse XF-24 extracellular flux analyzer (Seahorse Bioscience, North Billerica, MA, USA). On the day before the experiment, the sensor cartridge was placed into the calibration buffer supplied by Seahorse Bioscience and incubated at 37 °C in a non-CO_2_ incubator. Podocytes were cultured on Seahorse XF-24 plates at a density of 20,000 cells per well. The cells were washed and incubated with assay medium (DMEM without bicarbonate) at 37 °C in a non-CO_2_ incubator for 1 h. All media and injection reagents were adjusted to pH 7.4 on the day of the assay. Three baseline measurements of the OCR were collected before sequential injection of mitochondrial inhibitors. Three readings were taken after the addition of each mitochondrial inhibitor (before injecting the next inhibitor). The mitochondrial inhibitors used were oligomycin (2 μg/mL), CCCP (10 μM) and rotenone (1 μM). The OCR was automatically calculated and recorded by the Seahorse XF-24 software (Seahorse Bioscience, North Billerica, MA, USA). The percentage change compared with basal rates was calculated as the value of the measurement divided by the average value of the baseline readings.

### 4.8. Quantitative RT-PCR

Total RNA was isolated using the NucleoSpin^®®^ RNA II kit (Macherey-Nagal, Düren, Germany) [15]. Complementary DNA was prepared using M-MLV Reverse Transcriptase and oligo-dT primers (Invitrogen). To determine the efficacy of the CRIF1-specific siRNA, detection of *Crif1* mRNA levels was performed using a Rotor-Gene 6000 instrument (Corbett Life Science, Sydney, Australia) and QuantiTect™ SYBR^®®^ Green PCR Master Mix (Qiagen, Venlo, The Netherlands), according to the manufacturer’s instructions. The cycle number at which the fluorescence signal of a reaction crossed the threshold value was used as the basis for quantification, and data were analyzed using the Rotor-Gene™ 6000 real-time rotary analyzer software (version 1.7, Corbett Life Science, Sydney, Australia). The following primers were used: CRIF1 sense, 5′- TATCTCCTGCGGCTCTCTGT-3′; CRIF1 antisense, 5′-CTTCTGCTTTCGCCAGTTTT-3′. The *Crif1* expression levels were normalized to those of the mRNA encoding GAPDH, which was detected using commercially available QuantiTect primers (Qiagen).

### 4.9. Immunoblotting

Immunoblotting was performed according to standard methods using the following commercially available antibodies: anti-CRIF1 (sc-134882; Santa Cruz), anti-NDUFA9 (A21344; Molecular Probes, Eugene, OR, USA), anti-SDHA (A11142; Molecular Probes), anti-UQCRC2 (A11143; Molecular Probes), anti-COX1 (A6403; Molecular Probes), anti-ATP5A1 (A21350, Molecular Probes), anti-actin (sc-1616, Santa Cruz), anti-F-actin (ab205, Abcam, Cambridge, UK), anti-α-actinin-4 (sc-134236, Santa Cruz), anti-synaptopodin (sc-21537, Santa Cruz), anti-cofilin (sc-33779, Santa Cruz), anti-ZO-1 (sc-10804, Santa Cruz), and anti-nephrin (sc-32532, Santa Cruz).

### 4.10. Statistical Analysis

Data are represented as the mean ± SD. Statistical analyses were performed using Student’s *t*-tests. *p* < 0.05 was considered statistically significant.

## 5. Conclusions

In conclusion, our study shows that knockout of CRIF1 causes progressive albuminuria and segmental glomerulosclerosis in mouse kidney. The deletion of CRIF1 may be a useful tool for studying the primary role of mitochondria and cytoskeleton in podocytes.

## Figures and Tables

**Figure 1 ijms-22-04827-f001:**
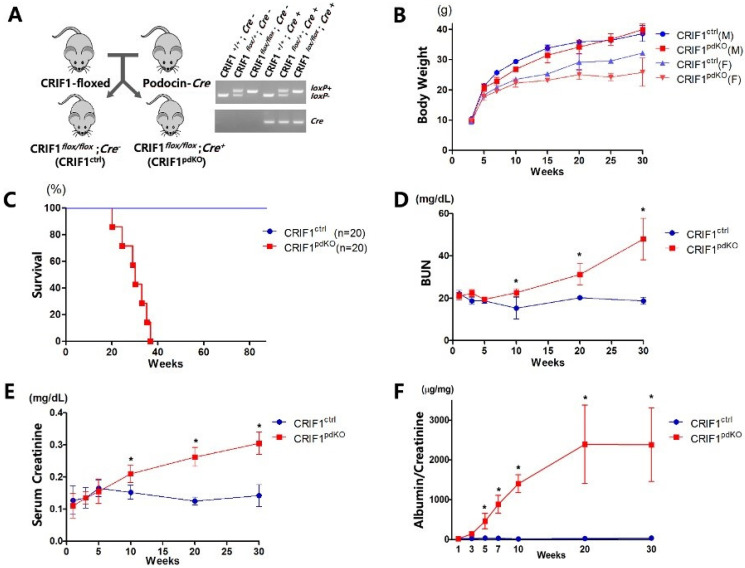
Podocyte-specific CRIF1 deficiency leads to massive albuminuria and azotemia in mice. (**A**) Podocyte-specific CRIF1 knockout mice were generated by crossing CRIF1^flox/flox^ mice with podocin-Cre mice. PCR analyses were performed to genotype the CRIF1-loxP alleles and Cre transgene. (**B**) Body weights of the male (M; *n* = 10) and female (F; *n* = 10) CRIF1^pdKO^ and CRIF1^ctrl^ mice, respectively. (**C**) Survival rates of the CRIF1^pdKO^ and CRIF1^ctrl^ mice (*n* = 20 per group). (**D**,**E**) Blood urea nitrogen (**D**) and serum creatinine (**E**) levels in the CRIF1^pdKO^ and CRIF1^ctrl^ mice (**F**) albuminuria levels in the CRIF1^pdKO^ and CRIF1^ctrl^ mice (**B**,**D**–**F**) Data are represented as the mean ± SD. (**D**–**F**) * *p* < 0.05 by Student’s *t*-tests.

**Figure 2 ijms-22-04827-f002:**
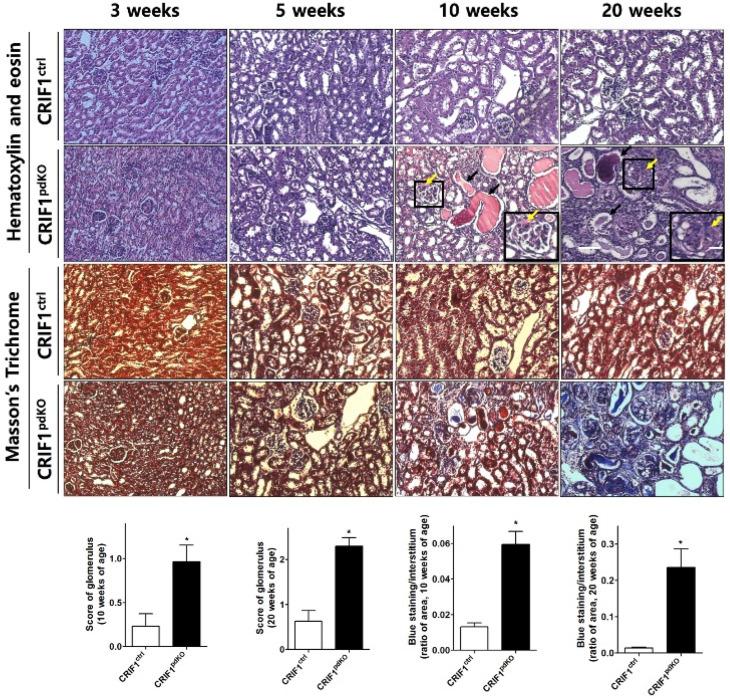
Podocyte-specific CRIF1 deficiency leads to glomerular sclerosis and tubulointerstitial fibrosis in mice. Hematoxylin and eosin staining (upper panels) and Masson’s Trichrome staining (lower panels) of the renal cortex in CRIF1^pdKO^ and CRIF1^ctrl^ mice at 3, 5, 10, and 20 weeks of age. Large black boxes show higher magnifications of the small black boxed areas in the larger panels. Scale bars, 20 μm (large panels) and 7 μm (small panels). The yellow and black arrows in the panels showing sections from 10- and 20-week-old CRIF1^pdKO^ mice indicate the occurrence of glomerular sclerosis and proteinaceous casts, respectively. The glomerular sclerosis scores and interstitial fibrosis areas in kidneys from 10- and 20-week-old CRIF1^ctrl^ and CRIF1^pdKO^ mice. Data are represented as the mean ± SD. * *p* < 0.05 by Student’s *t*-tests.

**Figure 3 ijms-22-04827-f003:**
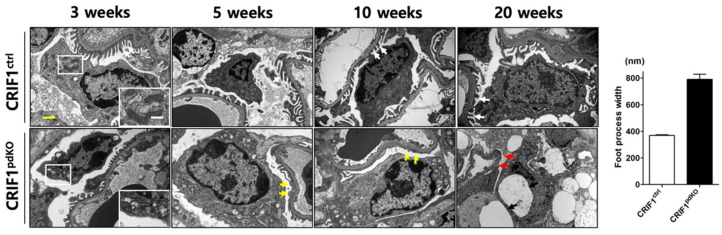
Transmission electron micrographs of renal sections from CRIF1^ctrl^ and CRIF1^pdKO^ mice at 3, 5, 10, and 20 weeks of age, and the foot process width of mice kidney at 5 weeks of age. White arrows in the panels showing 3-, 5-, 10-, and 20-week-old mice indicate foot process with normal structure. White boxes in the panel showing section from 3-week-old mice indicate mitochondria. The yellow arrows in the panel showing section from 5-week-old and 10-week-old CRIF1^pdKO^ mice indicate foot process fusion and effacement, and microvillous transformation. The red and black arrows in the panel showing a section from a 20-week-old CRIF1^pdKO^ mouse indicate extensive effacement and microvillous transformation of the foot processes, and podocyte degeneration and vacuolization, respectively. Yellow scale bar, 2 μm. White scale bar, 1 μm.

**Figure 4 ijms-22-04827-f004:**
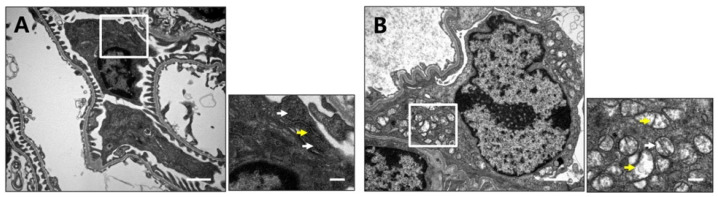
Podocyte-specific CRIF1 deficiency causes mitochondrial structural abnormalities. Transmission electron micrographs of renal cortex sections from 10-week-old CRIF1^ctrl^ mice (**A**) and CRIF1^pdKO^ (**B**) mice. The smaller panels show higher magnifications of the boxed areas in the larger panels. The white arrows indicate mitochondria. The yellow arrows indicate cristae of mitochondria Scale bars, 2 μm (large panels) and 0.5 μm (small panels).

**Figure 5 ijms-22-04827-f005:**
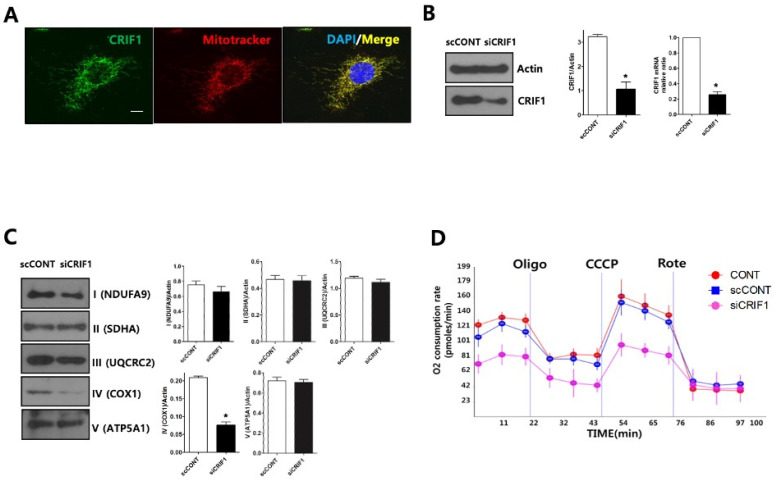
Deletion of CRIF1 causes mitochondrial dysfunction in podocytes. (**A**) Immunofluorescent staining of endogenous CRIF1 in an immortalized mouse podocyte. MitoTracker Red and DAPI were used to stain the mitochondria and nucleus, respectively. Scale bar, 20 μm. (**B**) Immunoblot analysis (left and middle panels) of CRIF1 protein expression and qRT-PCR analysis of *Crif1* mRNA expression in podocytes transfected with a scrambled control siRNA (scCONT) or a CRIF1-specific siRNA (siCRIF1). The expression levels of CRIF1 protein and *Crif1* mRNA were normalized to those of actin and GAPDH, respectively. Data are represented as the mean ± SD of *n* = triple replicates. (**C**) Immunoblot analyses of mitochondrial protein complexes I–IV. The expression levels of the complexes were normalized to those of actin. Data are represented as the mean ± SD of *n* = triple replicates. (**D**) The OCRs in non-transfected (CONT), control siRNA-transfected (scCONT), and CRIF1-specific siRNA-transfected podocytes after the addition of oligomycin (Oligo; 2 μg/mL), CCCP (10 μM) and rotenone (Rote; 1 μM). * *p* < 0.05 (**B**,**C**) by Student’s *t*-tests.

**Figure 6 ijms-22-04827-f006:**
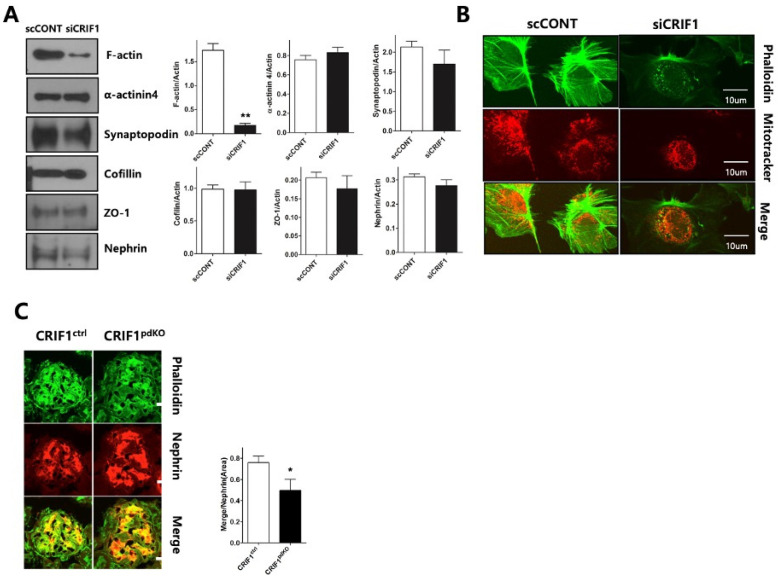
Deletion of CRIF1 causes loss and aggregation of F-actin in podocytes. (**A**) Immunoblot analyses of F-actin, α–actinin-4, synaptopodin, cofilin, ZO-1, and nephrin in control siRNA- and CIRF1-specific siRNA-transfected podocytes. Data are represented as the mean ± SD of *n* = triple replicates. (**B**) Immunofluorescent staining of phalloidin (F-actin) in control siRNA- and CIRF1-specific siRNA-transfected podocytes. Mitochondria were stained using MitoTracker Red. (**C**) Immunofluorescent staining of phalloidin (F-actin) and nephrin in kidneys from 10-week-old CRIF1^ctrl^ and CRIF1^pdKO^ mice. * *p* < 0.05, ** *p* < 0.01 by Student’s *t*-tests.

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
