# Peer review of "Mitochondrial Dysfunction in Podocytes Caused by CRIF1 Deficiency Leads to Progressive Albuminuria and Glomerular Sclerosis in Mice"

_ijms, 2021, doi:10.3390/ijms22094827_

Round 1

Reviewer 1 Report

Current study designed to study glomerular function of the novel podocyte-specific primary OxPhos dysfunction model mice. Researchers were able firstly to demonstrate the role of mitochondria in the maintenance of podocyte structure and function.  All results are described and illustrated perfectly.

Reviewer 2 Report

Na et al. studied changes in podocyte function in the genetic silencing mice of CRIF1 as well as in the cultured mouse podocytes. They showed that CRIF1, a regulatory factor of mitoribosome, silence contributes to mitochondrial dysfunction in podocytes and leads to glomerular disease. These in vivo observations were further confirmed in the cultured podocytes showing that less mitochondrial protein complex IV and oxygen consumption with a low expression of F-actin in the CRIF1-silencing cells.

These results are interesting. However, there are some concerns as listed below that authors should address.

  1. Please put all supplementary results back to the main figures because the results in S1-S4 figures are all significant to this study.
  2. In the abstract, please full spelling of CRIF1.
  3. Please provide the data about serum protein levels and liver function in mice because this may rule out overflow proteinuria seen in CRIF1pdKO is due to inadequate liver function.
  4. I suggest correct proteinuria to albuminuria because you use a specific ELISA kit to quantitate albumin.
  5. Please provide the data of creatinine clearance in Figure 1 because this will be more clear than gradual increases in serum BUN and creatinine to show changes in glomerular filtration.
  6. “e” was missing in the y-axis title of Figure 1F.
  7. In Figure 2, why there is a duplicate in histological images? Is that tissue sections from male and female mouse, respectively?
  8. In Figure 2, is there any data about the size of glomeruli because there is glomerular enlargement in the CRIF1pdKO kidney (similar to S4 Figure)?
  9. The legend title in Figure 2 is wrong, please check.
  10. Lines 107-110, these descriptions are for Figure 4 not for Figure 3.
  11. In Figure 3, why there is no yellow or black arrow in the image of 10 weeks? I suggest to mark normal foot process in the image of control mice.
  12. Line 217, please correct the section title of 2.4.
  13. In Figure 4A, why there is no yellow arrow to point cristae in the smaller panel of the control mice? Cristae is also present in the morphology of normal mitochondria, right?
  14. In Figure 5C, the representative blot of mitochondrial protein complex I clearly shows a smaller band in CRIF1pdKO than CRIF1ctrl but the means in the right statistic bar are similar.
  15. In Figure 5D, is there no statistical significance between siCRIF1 and controls?
  16. Please correct the legend title in Figure 5.
  17. What is n for Figure 6A?
  18. Please show the phase-contrast images of Figure 6B because smaller cell showing a less F-actin expression of course.
  19. In the IF staining of S4 Figure, the green (F-actin), red (nephrin), and yellow (merged) color areas in CRIF1pdKO are all larger than CRIF1ctrl. This does not match statistic result of the right bar graph. Please explain.
  20. Please discuss more about the novel role of CRIF1 in podocytes because CRIF1seems not only affect intracellular F-actin but have an influence on extracellular nephrin expression (S4 Figure).
